# Emerging Role of Phospholipase-Derived Cleavage Products in Regulating Eosinophil Activity: Focus on Lysophospholipids, Polyunsaturated Fatty Acids and Eicosanoids

**DOI:** 10.3390/ijms22094356

**Published:** 2021-04-21

**Authors:** Eva Knuplez, Eva Maria Sturm, Gunther Marsche

**Affiliations:** Otto Loewi Research Center, Division of Pharmacology, Medical University of Graz, 8010 Graz, Austria; eva.knuplez@medunigraz.at (E.K.); eva.sturm@medunigraz.at (E.M.S.)

**Keywords:** eosinophils, phospholipase, lipid mediators, arachidonic acid, eicosanoids, prostaglandins, leukotrienes, lysophospholipids, endocannabinoids

## Abstract

Eosinophils are important effector cells involved in allergic inflammation. When stimulated, eosinophils release a variety of mediators initiating, propagating, and maintaining local inflammation. Both, the activity and concentration of secreted and cytosolic phospholipases (PLAs) are increased in allergic inflammation, promoting the cleavage of phospholipids and thus the production of reactive lipid mediators. Eosinophils express high levels of secreted phospholipase A2 compared to other leukocytes, indicating their direct involvement in the production of lipid mediators during allergic inflammation. On the other side, eosinophils have also been recognized as crucial mediators with regulatory and homeostatic roles in local immunity and repair. Thus, targeting the complex network of lipid mediators offer a unique opportunity to target the over-activation and ‘pro-inflammatory’ phenotype of eosinophils without compromising the survival and functions of tissue-resident and homeostatic eosinophils. Here we provide a comprehensive overview of the critical role of phospholipase-derived lipid mediators in modulating eosinophil activity in health and disease. We focus on lysophospholipids, polyunsaturated fatty acids, and eicosanoids with exciting new perspectives for future drug development.

## 1. Introduction

Eosinophils are multifunctional leukocytes involved in the host defense against helminth infections, tissue homeostasis, and tissue repair [1,2,3]. Eosinophils are capable of releasing a variety of mediators upon activation that both promote and maintain local inflammation and contribute to the pathogenesis of allergic disease [3,4]. Eosinophils contain secondary granules consisting of cytotoxic major basic protein, eosinophil-derived neurotoxin, eosinophil peroxidase, eosinophil cationic protein, and transforming growth factor beta (TGF-β), all of which contribute to airway remodeling [5]. The effector functions of eosinophils promote airway epithelial damage and hyperreactivity, and thereby contribute to all major hallmarks of asthma pathophysiology [6]. Therefore, eosinophil tissue recruitment and activation have long been considered as potential targets for the discovery and development of anti-allergy drugs.

Recent breakthroughs in eosinophil targeting have been made in the field of biologics and have been discussed in detail elsewhere [7]. The application of monoclonal antibodies, such as benralizumab (targeting the interleukin (IL)-5 receptor) or lirentelimab (anti-Siglec-8 antibody), results in almost complete eosinophil elimination from blood and tissues, improving the patient’s clinical score and symptoms [8,9].

Recently, however, it has been recognized that eosinophils are crucial for local immunity and repair (LIAR hypothesis), by promoting immunoregulation and tissue homeostasis [1,2,10]. An important function of eosinophils is their antitumor effect in colorectal cancer [10,11,12]. For example, using a high-throughput evaluation of human primary tumors, Grisaru-Tal et al. identified differential infiltration of eosinophils in anatomically distinct tumors, with high eosinophil counts in mucosal barrier organs such as the colon or esophagus [13]. Eosinophils additionally show hepatoprotective activity [14] and cardiac protective function after myocardial infarction [15]. Of particular interest, a robust inverse correlation between eosinophil numbers and severity of novel coronavirus disease of 2019 (COVID-19) infection was observed [16,17,18,19,20]. Furthermore pre-existing eosinophilia in asthmatics was proven to be protective against COVID-19-associated hospital admission, while development of eosinophilia during hospitalization was associated with decreased mortality [21]. Taken together, these new findings call into question the long-term effects of eosinophil-depleting antibodies [22] and point to an unmet need of targeting eosinophil overactivation without completely depleting this multifunctional immune cell type.

It has long been known that both the concentration and activity of secreted and cytosolic phospholipases are increased in allergy and allergic inflammation, leading to increased phospholipid cleavage and lipid mediator production [23,24]. In fact, eosinophils have been discovered to express high levels of secreted phospholipase A_2_ in comparison to other leukocytes, indicating their involvement and contribution to lipid mediator production during allergic inflammation [25]. Recently, dysregulated fatty acid metabolism was identified in nasal polyp-derived eosinophils from patients with chronic rhinosinusitis [26], asthma [27], and other eosinophilic allergic diseases [28]. Therefore, targeting the complex network of lipid mediators has the potential to suppress eosinophil over-activation in allergy.

In this comprehensive review, we highlight the emerging role of phospholipases in regulating eosinophil activity, with a focus on reactive phospholipase-derived cleavage products, including lysophospholipids, polyunsaturated fatty acids, and eicosanoids.

## 2. Phospholipases Play a Key Role in Allergies

Phospholipases are enzymes that generate bioactive cleavage products, such as lysophospholipids and free fatty acids by cleaving membrane phospholipids. They can be broadly divided into two main groups depending on the site of phospholipid molecule cleavage [29]. Phospholipases A and B are acylhydrolases, hydrolyzing ester bonds at sn-1 and sn-2 position of the phospholipid molecule, while phospholipases C and D are phosphodiesterases [30]. Generally, phospholipases differ in their tissue expression patterns, cellular location, as well as substrate specificity [29]. Due to their ability to produce bioactive lipid mediators, they have long been considered as important enzymes involved in the pathogenesis of allergic inflammation [31].

### 2.1. Phospholipase A_2_ (PLA_2_) and Eosinophils

Phospholipases A_2_ are rate limiting enzymes cleaving phospholipids at the sn-2 position and thereby producing a free fatty acid and a lysophospholipid. The best described PLA_2_-derived free fatty acid is arachidonic acid—a precursor for the diverse family of eicosanoids [32]. The family of PLA_2_ comprises secreted phospholipases A_2_ (sPLA_2_), cytosolic phospholipases A_2_ (cPLA_2_), Ca^2+^-independent phospholipases A_2_ (iPLA_2_), platelet activating factor acetylhydrolases (PAF-AH), and lysosomal PLA_2_ (LPLA_2_) [33]. Interestingly, endogenous PLA_2_ activity was shown to regulate human eosinophil degranulation, therefore contributing to their cytotoxic functions [34].

#### 2.1.1. Secreted Phospholipases A_2_

Levels of secreted phospholipases A_2_ (sPLA_2_) have been found to be increased in the lungs and bronchoalveolar lavage fluid (BALF) of asthmatics following allergen challenge [24,35]. Moreover, analysis of the plasma metabolome discovered increased activity of sPLA_2_ in the lungs of guinea pigs in eosinophilic inflammation, while the levels of its substrate phosphatidylcholine were found to be decreased [36].

Recently, a specific subset of sPLA_2_–sPLA_2_ group X was identified to play a crucial role in allergic inflammation and airway responses to inhaled allergens [37,38,39]. Nolin et al. described that genetic deficiency of sPLA_2_-X results in a significant reduction in airway hyperresponsiveness as well as eosinophil trafficking into the airways [37]. Interestingly, endogenous expression of sPLA_2_-X has been described in eosinophils isolated from asthmatic patients. Endogenous sPLA_2_-X in eosinophils can translocate to granules and lipid bodies upon eosinophil activation and induce eicosanoid production through cytosolic PLA_2_ activation [40]. Similar mechanism of action and effects are observed, when exogenous sPLA_2_-X is added to eosinophils [41]. Importantly, targeting of sPLA_2_-X with specific small-molecule inhibitors such as RO061606 or ROC-0929 decreases eicosanoid release, allergen-induced airway inflammation, mucus hypersecretion, and hyperresponsiveness in preclinical studies (Table 1) [40,42]. Another subtype, sPLA2 group V-has recently been implicated in allergic inflammation [43,44]. It has been shown that macrophage-associated sPLA_2_-V activates lung type 2 innate lymphoid cells (ILC2s) and leads to lung eosinophilia through the increase of IL-33 [45]. Even though eosinophils were not found to express sPLA_2_-V themselves, it was demonstrated that exogenously added human sPLA_2_-V can activate eosinophils, inducing the liberation of arachidonic acid and eicosanoid production [46].

#### 2.1.2. Cytosolic Phospholipases A_2_

Cytosolic phospholipases A_2_ have long been recognized as vital for eicosanoid production, since inherited cPLA_2_ deficiency leads to widespread decreases in eicosanoid levels in humans [81]. Furthermore, increased expression of group IV cPLA2 has been identified in sputum cells from subjects with asthma and exercise-induced bronchoconstriction [82]. The involvement of cPLA_2_ -IV in eosinophil over-activation have been demonstrated in preclinical models, and cPLA_2_ inhibition prevented both eosinophilic infiltration and subsequent airway hyperresponsiveness after antigen challenge [47]. Moreover, clinical data from patients suffering from hyper-eosinophilic syndrome suggests that increased and activated (phosphorylated and membrane-translocated) cPLA_2_-IV is involved in the augmented release of leukotriene C4 from isolated eosinophils [83].

Drug development of specific cPLA_2_ inhibitors for treatment of asthma and allergic diseases has been underway for the past two decades, with many orally-bioavailable compounds entering human clinical trials [84,85,86] (Figure 1). For example, a topical solution of the cPLA_2_ inhibitor ZPL-521 has now completed a phase IIb clinical trial for treatment of atopic dermatitis [48] (NCT02795832). However, a major drawback of previously developed small inhibitors targeting cPLA_2_ is their high lipophilicity, which leads to unfavorable absorption, distribution, metabolism, and excretion [85]. Lately, a novel class of highly potent cPLA_2_-IV oxo-ester inhibitors with a favorable lipophilic profile was developed and characterized by Kokotou et al. [49]. These compounds (GK452, GK504, GK484) potently inhibit prostaglandin D2 production, but remain to be tested on eosinophils [49,50].

#### 2.1.3. Platelet Activating Factor Acetylhydrolase (PAF-AH)

PAF-AH hydrolyses the acetyl ester at the sn-2 position of platelet activating factor (PAF), thus producing lyso-PAF (alkyl-lysophosphatidylcholine) [87]. A minor fraction of PAF is synthesized de novo, while the main fraction of PAF is formed by the enzyme lysophosphatidylcholine acetyl-transferase (LPCAT) [88]. Both, pro-inflammatory and pro-allergic actions of PAF have been extensively studied as described by Gill et al. in detail [89]. Eosinophils, in particular, were found to produce PAF in response to chemotactic stimuli [89]. PAF was found to potently induce eosinophil degranulation in a non-receptor mediated fashion [90]. It was previously demonstrated that asthmatics have reduced levels of PAF-AH and application of recombinant PAF-AH showed great therapeutic potential in animal models of asthma [51]. However, a clinical trial investigating the use of recombinant human PAF-AH in mild asthmatics showed no improvement in clinical symptoms [52].

PAF was shown to act on a specific G protein-coupled receptor, inducing eosinophil activation, calcium flux, and prostanoid secretion [91,92]. Furthermore, PAF is a potent chemotactic and chemokinetic factor for eosinophils [93]. The effects of PAF and other phospholipid cleavage products on eosinophils have been summarized in Table 2. Currently, the orally bioavailable dual antagonist rupatadine (antagonist of the H1 histamine receptor and PAF-receptor) is indicated for use in urticaria and allergic rhinitis [94,95]. When compared with levocetirizine, rupatadine use resulted in less adverse effects as well as in significantly greater reductions in the total nasal symptoms score, serum immunoglobulin E (IgE) level as well as eosinophil counts in seasonal allergic rhinitis patients [53]. Moreover, studies conducted in pediatric allergic rhinitis patients demonstrated an acceptable safety profile and efficacy of rupatadine [96,97]. Rupatadine is recommended in current international guidelines as first-line therapy for the treatment of allergic rhinitis in children [98,99,100].

#### 2.1.4. Phospholipase A2 Cleavage Products: Lysophospholipids

The most common lysophospholipid produced by the action of phospholipases is lysophosphatidylcholine (LPC), which can reach plasma concentrations of up to 150 µM already under physiological conditions [130]. LPC is an amphiphilic molecule possessing a long aliphatic chain and a polar head giving it surfactant-like properties. Under its critical micellar concentration, LPC favors the bending of the cell membrane to a positive curvature, thereby altering its properties [131]. If LPC is used in high concentrations or in the absence of carrier proteins (e.g., albumin), it induces cell membrane lysis and cell death [132]. The concentration of LPC increases under inflammatory conditions, with increased levels of LPC being observed in BALF of asthmatic patients [130,133]. Conversely, a novel metabolomics-based approach identified decreased levels of LPC species in patients suffering from asthma [134].

Studies investigating the effects of LPC on immune cells are contradictory [135]. Specifically, most older studies investigating the effects of LPCs on eosinophils found them to be pro-inflammatory–increasing the activation, adhesion, and migration of eosinophils [115,116]. However, in these studies, the authors used LPC in the absence of physiological carrier proteins, which explains the toxic effects induced by LPC on eosinophils [115]. Importantly, Marathe et al. identified that many commercially available LPC species contain PAF or PAF-like contaminants, which are responsible for the observed eosinophil-chemotactic potential [136]. Moreover, when LPC was investigated in animal models of allergy and asthma, prolonged LPC application was often used [116,137]. Since LPC is short-lived and rapidly metabolized in tissues by the action of lysophospholipases (generating pro-inflammatory lysophosphatidic acid or used as a precursor for phosphatidylcholine synthesis), it is therefore not possible to determine whether the observed pro-inflammatory effects are due to the action of LPC or its metabolites [138,139].

In fact, Kwatia et al. reported that a combined action of both secretory phospholipases and eosinophil-derived lysophospholipases is needed to induce a loss in surfactant activity as observed in asthma [140]. Moreover, the Charcot Leyden-Crystal (CLC) or galectin-10, which is one of the most abundant proteins in human eosinophils, has been reported to cleave palmitate from LPC 16:0 [141,142]. Interestingly, levels of CLC in sputum serve as robust biomarkers of eosinophilic inflammation and were recently reported to act as a type 2 adjuvant–stimulating adaptive and innate immunity [143,144]. When anti-CLC antibodies were applied to patient-derived mucus, they were capable of dissolving patient-derived CLCs. Furthermore, anti-CLC antibodies reversed goblet-cell metaplasia, IgE synthesis as well as bronchial hyperreactivity in a humanized mouse model of asthma [144]. Overall, these studies suggest that LPC itself is not responsible for the observed proinflammatory effects on eosinophils.

Saturated LPC species added to human eosinophils in the presence of the physiological carrier, serum albumin, strongly suppressed upregulation of CD11b and degranulation [101]. The anti-inflammatory effects of LPC appear to be mainly mediated by its action on cell membranes, where it disrupts lipid raft formation. This leads to suppression of receptor activation and downstream signaling effects, such as Ca^2+^ flux and kinase phosphorylation [101]. Short-term application of LPC in mice inhibited the infiltration of eosinophils into BALF [101]. Interestingly, LPC has been shown to act as a non-competitive product inhibitor of secreted PLA_2_, which may be another anti-inflammatory mechanism [145].

Consistent with the anti-inflammatory effects observed with LPC [101], the stable LPC analogue miltefosine showed very similar actions. Miltefosine is an orally bioavailable alkylphosphocholine drug approved for the treatment of leishmaniasis [55]. Similarly to LPC, miltefosine inhibited human eosinophil responses in vitro [55]. Moreover, in an ovalbumin model of allergic inflammation, miltefosine suppressed eosinophil infiltration into BAL fluid, ameliorated allergic inflammation in vivo and led to an improvement in lung function parameters [55]. Therefore, highlighting the potential of miltefosine as a novel treatment option dampening the over-activation of eosinophils.

Recently, another lysophospholipid has been investigated in the context of eosinophil activation. Hwang et al. demonstrated that lyosphosphatidylserine is capable of inducing eosinophil degranulation via its action on P2Y_10_ receptor [102]. A subsequent study demonstrated the ability of lysophosphatidylserine to induce eosinophil extracellular trap formation [117]. Because P2Y_10_ is highly expressed during late stage eosinophil differentiation and in eosinophils isolated from severe asthmatics, it offers the possibility of targeting eosinophil degranulation without affecting their survival [102,117].

#### 2.1.5. Phospholipase A_2_ Cleavage Products: Free Fatty Acids

The most recognized polyunsaturated fatty acid (PUFA) produced by the action of phospholipases is ω-6 acid arachidonic acid (AA, 20:4) (Figure 2). Arachidonic acid can be metabolized by the action of cyclooxygenases (COX), lipoxygenases (LOX) and cytochrome P450 monoxygenases (CYP), leading to multiple bioactive mediator formation. COX enzymes produce prostaglandins and thromboxanes, LOX enzymes are responsible for the formation of leukotrienes and lipoxins while the CYP pathway generates hydroxy-eicosatetraenoic acids (HETEs) and epoxy-eicosatrienoic acids (EETs) [146]. The products of AA metabolism and their targeting are described in detail in Section 3.

Other important PUFAs involved in inflammation and allergy are the ω-3 acids eicosapentaenoic acid (EPA, 20:5) and docosahexaenoic acid (DHA, 22:6) (Figure 2). The supplementation of these ω-3 acids with diet was previously associated with favorable outcomes in a variety of disease models [147,148,149,150,151]. Both, EPA and DHA supplementation was discovered to increase membrane fluidity and lipid raft clustering leading to decreased cell activation [152]. In addition, EPA and DHA were found to compete with AA as a substrate for LOX, COX, and CYP enzymes. This reduces the formation of AA metabolites and replaces them with a series of specialized pro-resolving mediators (SPMs) [28,153].

## 3. Eicosanoids and Eosinophils

Eicosanoids are a diverse group of short-lived bioactive mediators mainly metabolized from arachidonic acid, which control a wide range of physiological processes such as vasoconstriction and vasodilatation, gastric motility, and bronchoconstriction. Cells of the innate immune system such as neutrophils, macrophages, mast cells, and eosinophils are capable of producing large amounts of eicosanoids, which may potentiate or attenuate local inflammatory processes. Moreover, eicosanoids were reported to play a vital role in a variety of diseases ranging from allergy to autoimmunity and cancer. Eicosanoids predominantly bind to and activate G protein-coupled receptors (GPCRs) and therefore represent a readily targetable group of lipid mediators [154,155,156].

### 3.1. Specialized Pro-Resolving Mediators: Resolvins, Protectins, and Lipoxins

SPMs consist of E-series resolvins, D-series resolvins, protectins, maresins, as well as AA-derived lipoxins. This group of anti-inflammatory mediators inhibits polymorphonuclear leukocyte infiltration and promotes resolution of inflammation via macrophage clearance [157,158,159].

Eosinophils are a major source of resolvin E3 produced via the 12/15-LOX pathway, whereas neutrophils produce mainly resolvin E1 and resolvin E2 via the 5-LOX pathway [160]. Eosinophil-derived resolvin E3 was specifically found to inhibit the chemotaxis of polymorphonuclear cells [160]. Recently, Sato et al. described that resolvin E3 is capable of promoting the resolution of house dust mite-induced airway inflammation by regulating the expression of the leukotriene B4 receptor BLT_1_ and inhibiting IL-17 and IL-23 cytokine release. This results in decreased immune cell infiltration into the airways and improved lung function parameters [57]. Following this discovery, novel highly potent (in fg dose range) deoxy-resolvin E3 derivates were synthesized and characterized [58]. Interestingly, eosinophils isolated from healthy individuals were discovered to generate high levels of protectin D1 through the action of 15-LOX. Protectin D1 has been similarly described to inhibit eosinophil chemotaxis and adhesion molecule expression, and to reduce airway hyperreactivity and inflammation in a preclinical model of allergic airway inflammation [62,144,159]. Interestingly, a lipidomics-based approach identified dysregulated PUFA metabolism in eosinophils in severe asthma [62,144,159]. In particular, 15-LOX-derived pro-resolving mediators such as protectin D1 and lipoxin A4 (LXA_4_) were found to be decreased in severe asthma and associated with aspirin sensitivity in aspirin-exacerbated respiratory disease [28,161,162,163]. Finally, the levels of the LXA_4_ receptor are decreased in severe asthma [164]. Previously, multiple anti-inflammatory and anti-allergic properties in preclinical models have been attributed to a stable analogue of LXA_4_ [61].

Since then, clinical studies with LXA_4_ analogs have demonstrated safety and clinical efficacy as a topical solution for infant eczema (15(R/S)-methyl-lipoxin A(4)) [63] and for inhalation in asthmatic children with acute episode (5(S),6(R)-lipoxin A4 methyl ester and BML-111) [62].

### 3.2. Leukotrienes

Leukotrienes are produced via enzymatic lipid peroxidation from arachidonic acid by the action of lipoxygenases (LOX). First, diverse LOX (5-LO, 12-LO and 15-LO) species convert arachidonic acid into hydroperoxyeicosatetraenoic acids (HpETEs) and HETEs. 5-LO next converts the intermediate HpETE into a transient epoxide intermediate leukotriene A_4_ (LTA_4_). Finally, LTA_4_ can be further metabolized via three different pathways into cysteinyl leukotrienes (LTC_4_, LTD_4_, LTE_4_), LTB_4_ and into anti-inflammatory lipoxins (LXA_4_ and LXB_4_) [155,165].

Cysteinyl leukotrienes have been reported to bind to 5 different receptors with varying affinities–cysteinyl leukotriene receptors 1 and 2 (CysLT_1_, CysLT_2_), as well as GPR99, GPR17 and P2Y_12_ [165,166,167]. Eosinophils have been described to express CysLT_1_, CysLT_2_, GPR99 and P2Y_12_ [168,169]. However, even though GPR17 expression has not been confirmed on eosinophils yet, a recent report describes higher GPR17 expression in nasal polyp tissue in eosinophilic versus non-eosinophilic chronic rhinosinusitis, suggesting its involvement in eosinophil infiltration [170]. It has long been recognized that eosinophils are a major source of cysteinyl leukotrienes [167,169]. Binding of cysteinyl leukotrienes to CysLT_1_ on eosinophils was found to increase their transendothelial migration, superoxide generation, and degranulation [109,110]. Additionally, cysteinyl leukotrienes are capable of inducing secretion from cell-free eosinophilic granules by acting on the CysLT receptors expressed by the granules themselves [111].

LTB_4_ is produced from LTA_4_ by the action of LTA_4_ hydrolase [171]. In contrast to cysteinyl leukotrienes, only two different receptors have been identified for LTB_4_—the so called BLT_1_ and BLT_2_ [172,173]. Binding of LTB_4_ to its receptors has previously been demonstrated to result in potent chemotaxis of neutrophils and eosinophils [112,174]. Interestingly, a recent report by Pal et al. clearly describes that, even though LTB_4_ production has traditionally been attributed to neutrophils and alveolar macrophages, eosinophils are actually a major source of LTB_4_. The authors demonstrate that human eosinophils isolated from patients with severe asthma express the enzyme LTA_4_ hydrolase and produce LTB_4_ in response to stimuli [175].

Cysteinyl leukotrienes and LTB_4_ have long been implicated in allergy and T_H_2 type inflammation. Their role as potential therapeutic targets in allergic diseases has recently been comprehensively reviewed by Watanabe and colleagues [176]. In brief, the 5-LO inhibitor zileuton is currently registered for treatment of chronic asthma for patients of 12 years and older and additionally shows beneficial clinical improvement in upper airway inflammatory conditions and atopic dermatitis [70]. By decreasing the production of leukotrienes B_4_, C_4_, D_4_, and E_4_, zileuton reduces inflammation, edema, mucus secretion, and bronchoconstriction in the airways [70]. Further, montelukast, a specific antagonist of the CysLT_1_ receptor is routinely used for treatment of chronic asthma, prevention of exercise-induced bronchospasm, and the relief of allergic rhinitis symptoms [71,177]. Zafirlukast is another CysLT_1_ antagonist approved for the treatment of asthma in children 5 years of age and older, as well as off-label for allergic rhinitis and prophylaxis of exercise-induced bronchospasm. Both CysLT_1_ antagonists are administered orally and are preferred by children over inhaled corticosteroids [71]. Recently, montelukast additionally exhibited potential to reduce airway inflammation and hyperresponsiveness in a viral-triggered asthma exacerbation model [178].

While targeting the cysteinyl pathway resulted in successful clinical trials and drug development, the opposite can be said for targeting the LTB_4_/BLT_1_/BLT_2_ axis. While some earlier reports showed beneficial properties of a LTB_4_ antagonist in vitro and in preclinical models, the results could not be reproduced in a clinical patient setting [179,180,181]. This contradicting data might be explained by recent findings demonstrating anti-inflammatory actions of the BLT_2_ receptor, opposing the pro-inflammatory actions of the BLT_1_ receptor [182,183,184,185].

Eosinophils have been found to express high levels of 15-LOX and are capable of metabolizing 15-HPETE into 14,15-leukotrienes [186,187]. Since eosinophils were discovered to be a major source of these mediators, they were later renamed into eoxins [186,187]. Treatment of eosinophils with arachidonic acid leads to the formation of eoxin (EX) C_4_, which can be further metabolized into EXD_4_ and EXE_4_. Moreover, mast cells, human nasal polyps, and airway epithelial cells were also reported to release eoxins, which act by increasing vascular permeability and enhancing edema formation [186,187,188]. Both the levels of 15-LOX and EXC_4_ have been reported to be increased in asthma [189,190,191,192]. However, a separate study by Ono et al. found no correlation between eosinophil counts and EXC_4_ concentration in bronchoalveolar lavage fluid of patients with inflammatory lung diseases [193]. So far, a specific eoxin receptor has not been identified [187].

### 3.3. Prostaglandins

Prostaglandins are bioactive mediators formed from arachidonic acid via the cyclooxygenase pathway. First, an intermediate prostaglandin H_2_ (PGH_2_) is generated, which serves as a substrate for various synthases that form PGD_2_, PGE_2_, PGI_2_, PGF_2α_, and thromboxane [154]. Since the biology of prostaglandins and their targeting has been thoroughly reviewed recently by Lee et al. [194], this section provides only a cursory overview of previously published studies.

#### 3.3.1. Prostaglandin D_2_

In the peripheral tissues, PGD_2_ is metabolized from PGH_2_ via the action of hematopoietic PGD synthase (hPGDS) [195,196]. PGD_2_ is reported to bind to two receptors DP1 and DP2 (also called chemoattractant homologous molecule expressed on T_H_2 cells–CRTH2) [197]. Both, the hPGDS and the receptors DP1 and DP2 are expressed by eosinophils [132,198,199]. Hence, eosinophils are able to produce and release PGD_2_ following stimulation [200]. Furthermore, eosinophils isolated from patients with aspirin exacerbated respiratory disease produce more PGD_2_ and express higher levels of hPGDS [201]. Similarly, higher hPGDS expression was observed in tissue from allergic rhinitis patients [202]. Finally, in a study published recently, corticosteroid treatment was shown to downregulate hPGDS and DP2 expression in aspirin exacerbated respiratory disease [203].

The multiple pro-inflammatory actions of PGD_2_ on its receptors DP1 and DP2 on eosinophils have been reviewed in detail by Peinhaupt et al. [197]. Briefly, binding of PGD_2_ or selective agonists to DP2 promotes Ca^2+^ flux, CD11b upregulation, migration and degranulation of human eosinophils [118,119]. Moreover, by acting via DP2, PGD_2_ was found to prime eosinophils for other chemoattractants [120,121]. Activation of DP1 on eosinophils has in turn been studied in much less detail. A study by Gervais et al. showed that activation of DP1 on eosinophils prolongs their survival, whereas Peinhaupt et al. elucidated that the mechanism behind prolonged survival involves the induction of serum response element and pro-inflammatory genes [118,122]. Moreover, DP1 was found to enhance and modulate DP2 signaling and the co-operation of both receptors was shown to be crucial for efficient LTC_4_ secretion from eosinophils [123,204].

For targeting the PGD_2_ pathway in allergic inflammation, mainly DP2 antagonists have been used, while only a few reports studied the effects of hPGDS inhibition. In a recent preclinical study of allergic rhinitis, the hPGDS inhibitor TAS-205 in combination with montelukast showed a significant additive inhibitory effect on eosinophil infiltration and nasal obstruction in the late phase, when compared to treatment with either active agent alone [73]. Moreover, another orally-bioavailable hPGDS inhibitor (HQL-79) was found to reduce airway inflammation in an ovalbumin-model of allergic inflammation [72]. Currently, an ongoing phase I clinical study is assessing the safety and pharmacokinetics of the reversible hPGDS inhibitor ZL-2102 for treatment of chronic obstructive pulmonary disease, idiopathic pulmonary fibrosis, and asthma (NCT02397005).

In contrast, DP receptors have been studied in detail in multiple clinical studies with mixed results [202,205,206]. Fevipiprant (an oral antagonist of the DP2 receptor) was tested in two phase III clinical studies (LUSTER-1 and LUSTER-2), but neither of the two clinical trials demonstrated statistically significant reductions in asthma exacerbations [74]. Similarly, another DP2 antagonist AZD1981 recently showed no added benefit as an add-on therapy to inhaled corticosteroids and long-acting β2 agonists in atopic asthma [207]. However, promising results of the DP1 antagonist ONO-4053 for the treatment of allergic rhinitis were obtained in a phase II clinical study [75].

#### 3.3.2. Prostaglandin E_2_

PGE_2_ is produced from PGH_2_ by the action of prostaglandin E synthases [198]. While PGE_2_ (together with LXA_4_ and PGI_2_) is hypothesized to be an arachidonic acid-derived pro-resolving mediator [199], it also shows pro-inflammatory activities such as exacerbating autoimmune diseases [208]. Moreover, due to its immunosuppressive actions, PGE_2_ is considered detrimental in the tumor microenvironment [209,210,211]. On one hand levels of PGE_2_ were found to be increased in asthma and non-asthmatic eosinophilic bronchitis, while on the other hand, Aggarwal et al. reported an inverse relationship between PGE_2_ levels and eosinophil sputum counts [205,206,212]. PGE_2_ is reported to activate four receptors (EP1, EP2, EP3 and EP4). Of those, eosinophils were reported to express EP2 and EP4 receptors [78,213,214]. Of note, Durchschein et al. recently demonstrated a decreased expression of EP2 and EP4 receptors on blood eosinophils in a small cohort of patients with eosinophilic esophagitis [215].

Upon binding to EP2 and EP4 on eosinophils, PGE_2_ inhibits their effector responses such as chemotaxis and degranulation as well as CD11b upregulation [76,124]. Surprisingly, PGE_2_ was also found to promote survival of eosinophils [125,126]. By using an EP4 agonist, it was further demonstrated that the PGE_2_-EP4 axis is important for the interactions of eosinophils with endothelial cells [216]. Finally, the EP2 receptor agonist butaprost was shown to attenuate airway inflammation in a ovalbumin model of allergic airway inflammation [76], and in two recent studies investigating mast cell-mediated allergic responses [77,217]. In mild asthmatic patients, inhalation of PGE_2_ prior to allergen challenge significantly improved lung function parameters and attenuated airway inflammation and reduced eosinophil counts in sputum [127]. Although PGE_2_ suppresses eosinophil activation, PGE_2_ can activate other immune cell types and therefore may promote the development of diseases such as atherosclerosis, rheumatoid arthritis, cancer, and influenza virus infections [218,219,220,221,222]. Moreover, a recent study established a connection between PGE_2_-induced impaired immune response, male sex, and the risk of cardiovascular complications in COVID-19 patients [213]. Thus, application of this multifaceted lipid mediator or its analogues would most likely result in numerous side-effects.

#### 3.3.3. Prostaglandin I_2_-Prostacyclin

Prostacyclin (PGI_2_) is synthesized from PGH_2_ via the action of prostaglandin I synthase [214]. It is mainly produced in the vasculature, where it was found to be vasodilatatory and anti-aggregatory and is therefore considered an important player in cardiovascular health [214,223]. A stable inhaled analogue of PGI_2_–iloprost is currently indicated for treatment of pulmonary arterial hypertension [214,224]. Moreover, a recent multi-center clinical study investigated its beneficial effects as an add-on therapy in mechanically ventilated COVID-19 patients (NCT04420741) [225]. Although the beneficial effects of PGI_2_ on the vasculature have been thoroughly studied in the past, its immunomodulatory functions have only recently been recognized and are currently being studied in more detail [214].

Comparable to the effects of PGE_2_, PGI_2_ was also reported to inhibit eosinophil activation. Specifically, PGI_2_ as well as iloprost were discovered to inhibit the migration of guinea pig bone marrow eosinophils and human eosinophils via the PGI_2_ receptor IP [79,128]. Endothelium-derived PGI_2_ was found to modulate eosinophil–endothelial interaction and to affect eosinophil adhesion and transmigration, while promoting the barrier function in endothelial cells [128].

Overall, IP-receptor deficient mice exhibited exacerbated allergic inflammatory responses and airway remodeling, while inhalation of iloprost improved features of asthma in a preclinical model [80,226,227]. Recently, the use of IP-receptor deficient mice further elucidated the role of PGI_2_ in influencing the numbers and characteristics of natural killer cells, which limit lung innate lymphoid cells type 2 (ILC2s), preventing house dust mite-induced allergic inflammation [228].

### 3.4. 5-Oxo-8,11,14-Eicosatrienoic Acid (5-Oxo-ETE)

5-oxo-ETE is a lipid mediator produced from AA via oxidation of the 5-LO intermediate 5-HETE by the 5-hydroxyeicosanoid dehydrogenase (5-HEDH). 5-HETE has been shown to induce neutrophil and eosinophil migration and activation in vitro and in vivo and increased 5-HETE release has been observed in alveolar macrophages from asthmatic patients [229]. Compared to other eicosanoids, 5-HETE possesses only weak biological activity itself, but is further oxidized to 5-oxo-ETE, a potent chemoattractant for eosinophils and neutrophils. The precise biosynthesis pathway of 5-oxo-ETE has been reviewed in detail by Powell et al. [230]. It has been shown that concentration of 5-oxo-ETE is increased in exhaled breath condensates and correlates with the number of peripheral blood eosinophils following allergen challenge in patients sensitized to house dust mite [231]. Moreover, 5-oxo-ETE derived from nasal epithelial cells was shown to upregulate eosinophil cationic protein in eosinophils in nasal polyps [108]. Finally, an intradermal application of 5-oxo-ETE induced potent infiltration of both eosinophils and neutrophils into the skin, whereby infiltration of eosinophils was more than three times higher in asthmatic patients compared to control subjects [107].

5-HETE and 5-oxo-ETE were so far reported to bind to and activate the GPCR OXER1 (also known as TG1019) [232,233]. Expression of the receptor was detected on eosinophils, neutrophils, bronchoalveolar macrophages, basophils, monocytes, and a variety of cancer cell lines, with the highest expression on eosinophils [106,232,234]. Binding of 5-oxo-ETE to its receptor on eosinophils was reported to result in potent chemotaxis (highest among lipid mediators), increased cell activation, and degranulation [104,105,226,227,235,236]. Moreover, priming of eosinophils with granulocyte-macrophage colony-stimulating factor enhances 5-oxo-ETE-induced eosinophil degranulation [236]. 5-oxo-ETE can additionally induce the migration of eosinophils across endothelial cell layers and across the basement membrane by activating matrix metalloproteinases MMP-7 and MMP-9, which can degrade matrix components and ease the passage of eosinophils [237,238].

In contrast to research in the field of leukotrienes and prostaglandins, very few studies have been performed on 5-oxo-ETE and its receptor. This is mainly due to the absence of 5-oxo-ETE receptor and its orthologs in rodents limiting preclinical research. However, recently novel selective 5-oxo-receptor antagonists were characterized and tested [82,239,240]. Of those, S-Y048, which is highly potent and orally bioavailable, demonstrated a significant inhibition of eosinophil infiltration into the skin in response to intradermally administered 5-oxo-ETE and house dust mite in cynomolgus monkeys [80]. Thus, targeting of 5-oxo-ETE shows potential as a novel therapy for asthma and eosinophil-associated diseases [241].

## 4. Phospholipase C (PLC) Cleavage Products: Diacylglycerol (DAG) and Inositol 1,4,5-Trisphosphate (IP_3_) and Endocannabinoids

Phospholipase C hydrolyzes phosphatidylinositol 4,5-bisphosphate (PIP_2_) at the phosphodiester bond between the glycerol backbone and the phosphate group. PLC produces diacylglycerol (DAG) and inositol 1,4,5-trisphosphate (IP_3_) in response to cell activation signals [242]. Both IP_3_ and DAG are crucial molecules regulating signal transduction during cell activation and play a role in development of various diseases [243]. As such, PLC activation was shown to be involved in PAF-induced eosinophil degranulation and leukotriene secretion [244,245]. Synthetic alkyl ether lipids such as edelfosine and miltefosine are inhibitors of PLC activity. Following eosinophil stimulation, both alkyl ether lipids were shown to inhibit eosinophil granule release and Ca^2+^ flux, resulting from an increase in IP_3_ concentration [57,246,247].

In contrast to soluble IP_3_, DAG remains bound at the cell membrane, where diacylglycerol lipase may hydrolyze it to 2-arachidonolyglycerol (2-AG) [248]. Eosinophils were recently discovered to express high levels of DAG lipase in contrast to neutrophils and lymphocytes, indicating the importance of the pathway for 2-AG synthesis in these cells [249].

2-AG belongs to the endocannabinoid system consisting of endocannabinoid ligands, endocannabinoid receptors, and enzymes related to their synthesis and degradation. While many studies depict the roles of the endocannabinoid system in allergic diseases, their actual contribution remains controversial [250]. 2-AG binds to and activates GPCRs such as the cannabinoid receptor 1 and 2 (CB_1_, CB_2_). Moreover, 2-AG may be converted to arachidonic acid via the monoacylglycerol lipase (MGL) pathway leading to eicosanoid production [250]. Human leukocytes were found to differentially express CB_1_ and CB_2_ receptors as well as MGL [251,252,253]. Eosinophils, in particular, were shown to express MGL and CB_2_ receptor [117,251,253] and were found to migrate towards 2-AG [254,255]. Furthermore, the expression of the CB_2_ receptor is increased in eosinophils from symptomatic allergic donors and upregulated following ovalbumin challenge in mice, while CB_1_ expression was found to be upregulated in tonsils from allergic rhinitis patients [117,256,257]. In addition, their endogenous ligand 2-AG was found to be increased following challenge in both a contact dermatitis and dust mite antigen-induced dermatitis mouse model [64,258].

The CB_2_ receptor is known to mediate anti-inflammatory effects in many diseases, such as atherosclerosis and stress-induced neuroinflammation [239,240,259,260]. However, its activation on eosinophils surprisingly resulted in increased eosinophil reactivity and exacerbation of allergen-induced inflammation [103,258]. The CB_2_ receptor agonist S-777469 exhibited efficacy for treatment of atopic dermatitis in preclinical studies, supposedly via blockade of endogenous 2-AG binding to CB_2_. Unfortunately, results from a phase Ib/IIa clinical study in patients (NCT00697710) have not yet been made available [64].

Another possibility of targeting the 2-AG metabolism in eosinophils arises from specific blockade of its hydrolyzing enzyme MGL. A recent report by Abohalaka et al. suggested that inhibition of MGL with the specific inhibitor JZL184 ameliorates lipopolysaccharide-induced airway inflammation [65]. Similar findings were observed in human eosinophils in vitro, where MGL inhibition prevented the migration of primed eosinophils in response to 2-AG, suggesting potential anti-inflammatory actions of MGL inhibitors in lung inflammation and allergy [66]. The involvement of the endocannabinoid system in allergic diseases was comprehensively described by Angelina et al. in a recent Collegium Internationale Allergologicum update [250].

## 5. Phospholipase D Cleavage Product: Lysophosphatidic Acid (LPA)

Phospholipases D cleave the phospholipid molecule yielding the phosphatidic acid and the phospholipid headgroup (e.g., choline, ethanolamine…). Consequently, phosphatidic acid can be further metabolized into lysophosphatidic acid (LPA). Another pathway resulting in LPA production involves the hydrolysis of LPC via lysophospholipase D (also called autotaxin) [261]. Multiple studies suggest that both LPA concentration (specifically polyunsaturated species) and autotaxin levels and activity are increased in allergic diseases and asthma [67,261,262,263]. Moreover, serum autotaxin levels were found to correlate with pruritus in patients with atopic dermatitis [246]. Park et al. reported that autotaxin overexpressing mice exhibited an aggravated asthmatic phenotype, while blocking the enzymatic activity of autotaxin attenuated allergic inflammation [67]. A small molecule inhibitor of autotaxin (GLPG1690, ziritaxestat) was advanced to phase 3 clinical trials for idiopathic pulmonary fibrosis and phase 2 for systemic sclerosis, but all trials were discontinued in February 2021 (NCT03711162, NCT03976648) [247]. So far, autotaxin inhibition in allergy has not been tested in clinical trials.

Lysophosphatidic acid is reported to bind to and activate six different GPCRs (LPA_1–6_)—the so-called canonical LPA receptors [261,264]. The LPA receptor expression varies between immune cells, for instance eosinophils were reported to express LPA_1_ and LPA_3_ receptor, but lack LPA_2_ receptor expression [113,264]. Stimulation of human eosinophils with LPA resulted in chemotaxis, reactive oxygen production, Ca^2+^ flux, and CD11b upregulation [113]. Furthermore, in vivo application of LPA enhanced infiltration and action of guinea pig eosinophils [114]. Most studies investigating LPA receptor antagonism in allergic airway diseases focus on targeting LPA_2_, which reportedly suppresses airway hyperresponsiveness and immune cell infiltration in the BALF and lung tissue [67,251,264,265]. However, a recent study implicates the involvement of LPA receptors 1 and 3 in a neural mechanism involving LPA- induced carotid body activation, leading to acute bronchoconstriction [68]. Moreover, blockade of LPA receptors 1 and 3 (with the specific antagonist ki16425) reduced bradykinin-induced asthmatic bronchoconstriction and suppressed respiratory distress following allergen challenge [68].

Additionally, LPA is reported to activate non-canonical receptors such as peroxisome proliferator-activated receptor (PPAR)-γ, vanilloid receptor 1 channel as well as the receptor for advanced glycosylation products (RAGE) [252,253,256,257]. RAGE has been recently identified as a major mediator in inflammatory lung diseases, including asthma [266,267,268]. In fact, RAGE has been pinpointed as a critical mediator of T_H_2 signaling in the lung (using either small molecule inhibitors (e.g., FPS-ZM1) or mice lacking RAGE expression) [69,269]. In particular, application of the antagonist FPS-ZM1 drastically inhibited eosinophil infiltration following intranasal IL-5/IL-13 application [69]. RAGE is highly expressed on Type 1 alveolar epithelial cells as well as on immune cells such as eosinophils, promoting their survival [266,270,271,272]. Intriguingly, even though Dyer et al. demonstrated high RAGE expression on eosinophils, they could not observe receptor activation following activation with its natural ligand high-mobility group box 1 protein [272].

The role of autotaxin and LPA in immune regulation and asthma was recently reviewed by Kim et al. [252]. In their excellent review, the authors address an unmet need for available transgenic animal models to study cell-specific LPA production as well as LPA receptor expression and involvement in asthma and other immune diseases.

## 6. Non-Enzymatic Lipid Peroxidation

Exposure to allergic stimuli as well as gaseous pollutants, chemicals, viruses, and bacteria leads to the recruitment of eosinophils and neutrophils, which have a high capacity for reactive oxygen species (ROS) production. Activation of these inflammatory cells leads to the so-called ‘respiratory burst’, which is characterized by an increased production of O_2_·^−^,·OH, H_2_O_2_, hypochlorous and hypobromous acids [273,274,275]. This increased oxidative stress is associated with asthma pathophysiology, including airway hyperactivity and remodeling [276,277,278,279,280].

Eosinophils are reported to release up to two times more ROS compared to neutrophils [281]. Moreover, eosinophils express high levels of eosinophil peroxidase (EPO), which catalyzes many oxidative reactions and generates hypobromous acid capable of brominating protein tyrosine residues [282,283]. Furthermore, eosinophils from asthmatics were found to produce more ROS compared to healthy donors. [284,285] contributing to disease development through oxidation of DNA, proteins, and lipids [286]. Interestingly, ROS production is crucial for eosinophil extracellular trap release and the resulting airway damage [287].

Lipid peroxidation is generally considered as the main molecular mechanism involved in oxidative damage to cell structures. PUFAs, in particular, are prone to non-enzymatic lipid peroxidation due to the presence of the double bond near the methylene bridge, leading to autooxidation by the attack of free radicals [288]. Bioactive metabolites such as isoprostanes, dihomo-isoprostanes, isofurans, and neuroprostanes are formed by non-enzymatic PUFA peroxidation [289]. These mediators participate in many pathophysiological processes and are considered as biomarkers of oxidative stress [289,290,291]. Among these, especially stable F2-isoprostanes were found to be increased following allergen challenge in mice and humans as well as in asthmatic subjects [292,293,294,295,296,297].

Eosinophils contribute to oxidative stress in mild asthma and targeting oxidative stress could be considered as a treatment option during asthma exacerbations [298]. Preclinical data from Silveira et al. demonstrated that the ROS inhibitors N-acetylcysteine or diphenyleneiodonium reduced eosinophil extracellular trap formation by eosinophils and ameliorated airway inflammation [287]. Antioxidant supplementation with free radical scavengers, such as vitamin C or vitamin E has also been extensively studied in asthma and allergic diseases [299,300,301]. However, conflicting reports exist on the potential benefits of vitamin E supplementation, which acts as an inhibitor of the propagation cycle of lipid peroxidation [302,303,304]. This was partly explained by the report from Berdnikovs at al., which showed opposing functions of vitamin E isoforms on eosinophil and other immune cell recruitment in an experimental asthma model [305].

## 7. Conclusions

Phospholipases produce multiple reactive cleavage products, including lysophospholipids, polyunsaturated fatty acids, and eicosanoids. The highly complex network of lipids and lipid mediators modulates a variety of eosinophil functions in health and disease. The discovery of novel subclasses of lipid mediators and their receptors opens new possibilities for drug targeting. Because lipid mediator levels and the activity of their metabolizing enzymes are dysregulated in allergic inflammation and eosinophil-associated diseases, they offer a unique opportunity to target the over-activation and ‘pro-inflammatory’ phenotype of eosinophils. Importantly, this could be achieved without compromising the survival and functions of tissue-resident and homeostatic eosinophils. Fortunately, most lipid mediators function through GPCRs, which are highly drug-responsive receptors that can be readily targeted with orally bioavailable small molecule drugs.

## Figures and Tables

**Figure 1 ijms-22-04356-f001:**
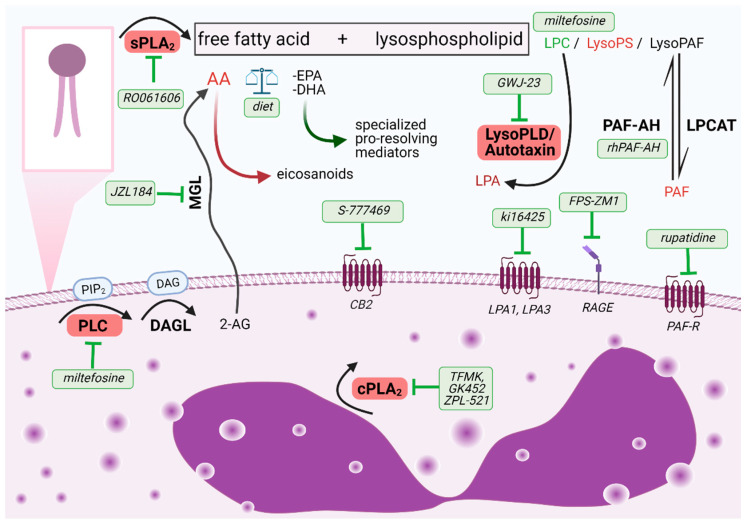
Overview of the phospholipid cleavage by phospholipases and major bioactive mediators produced during this process. Identified potential inhibitors of (i) enzyme activities or (ii) receptor antagonists targeting eosinophil over-activation are highlighted in green. Abbreviations: 2-AG, 2-arachidonoylglycerol; AA, arachidonic acid; CB2, cannabinoid receptor type 2, cPLA_2_, cytosolic phospholipase A2; DAG, diacylglycerol; DAGL, diacylglycerol lipase; DHA, docosahexaenoic acid; EPA, eicosapentaenoic acid; LPA, lysophosphatidic acid; LPA1,3, lysophosphatidic acid receptors type 1 and 3; LPC, lysophosphatidylcholine; LPCAT, lysophosphatidylcholine acyltransferase; LysoPAF, lyso platelet-activating factor; LysoPLD, lysophophospholipase D; LysoPS, lysophosphatidylserine; MGL, monoacylglycerol lipase; PAF, platelet-activating factor; PAF-AH, platelet-activating factor acetylhydrolase; PAF-R, platelet-activating factor receptor; PIP2, phosphatidylinositol 4,5-bisphosphate; PLC, phospholipase C; RAGE, receptor for advanced glycation end products; sPLA_2_, secreted phospholipase A2.

**Figure 2 ijms-22-04356-f002:**
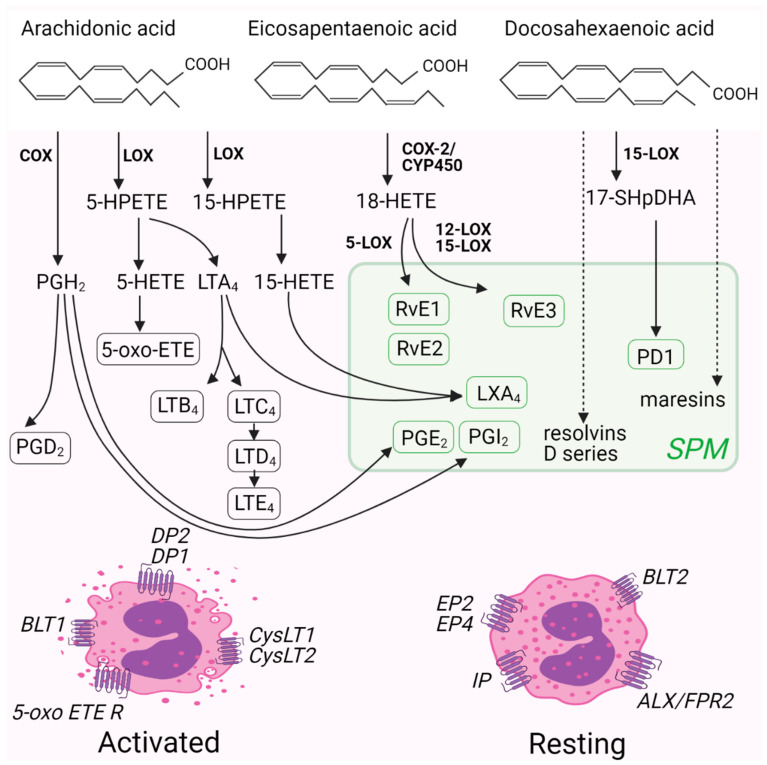
A schematic representation of PUVA metabolism into bioactive lipid mediators. PUVA are converted into bioactive lipid mediators capable of potentiating or inhibiting eosinophil activation via their specific G protein-coupled receptors. Abbreviations: 5-oxo-ETE, 5-oxo-eicosatetraenoic acid; 5-oxo-ETE R, 5-oxo-eicosatetraenoic acid receptor; ALX/FPR2, lipoxin A4 receptor/formyl peptide receptor 2; BLT 1 or 2, leukotriene B4 receptor 1 or 2; COX, cyclooxygenase; CYP450, cytochrome P450; CysLT1 or 2, cysteinyl leukotriene receptor 1 or 2; DP1 or 2, prostaglandin D2 receptor 1 or 2; EP2 or 4; prostaglandin E2 receptor 2 or 4; HETE, hydroxyeicosatetraenoic acid, HPETE, hydroperoxyeicosatetraenoic acid; HpDHA, hydroperoxydocosahexaenoic acid; IP, prostacyclin receptor; LOX, lipoxygenase; LTA4, leukotriene A4; LTB4, leukotriene B4; LTC4, leukotriene C4; LTD4, leukotriene D4; LTE4, leukotriene E4; LXA4, lipoxin A4; PD1, protectin D1; PGD2, prostaglandin D2; PGE2, prostaglandin E2; PGH2, prostaglandin H2; PGI2, prostacyclin; RvE1, resolvin E1; RvE2, resolvin E2; RvE3, resolvin E3; SPM, specialized pro-resolving mediators.

**Table 1 ijms-22-04356-t001:** Novel therapeutic options targeting lipid mediators with potential for targeting eosinophilic over-activation.

Compound	Mode of Action	Published Effects	References
RO061606 ROC-0929	inhibitor of sPLA_2_-X	Inhibition of eicosanoid formation. Decrease in allergen-induced airway inflammation, mucus hypersecretion and hyperresponsiveness.	[40,42]
trifluoromethyl ketone	inhibitor of cPLA_2_	Inhibition of antigen-induced airway eosinophil accumulation and airway hyperresponsiveness to methacholine.	[47]
ZPL-521	inhibitor of cPLA_2_	Clinical data: efficacious for treatment of atopic dermatitis:improvement on the eczema severity score, safe, well tolerated.	[48]
GK452, GK504, GK484	inhibitors of cPLA_2_ with improved ADME properties	Decreased eicosanoid production.	[49,50]
rhPAF-AH	hydrolysis of PAF to lysoPAF	Preclinical model: significant reduction in airway eosinophil infiltration, mucus hypersecretion, and airway hyperreactivity in response to methacholine challenge. Clinical data: No improvement in symptoms in mild asthma.	[51,52]
rupatadine	dual antagonist of H1 receptor and PAF receptor	In use for the treatment of urticaria and allergic rhinitis in adults and children. Reduces the levels of infiltrating eosinophils in allergic rhinitis.	[53,54]
miltefosine	lipid raft modulator, PLC inhibitor	Inhibits activation, degranulation, and migration of human eosinophils; decreases immune cell infiltration and improves lung function parameters in a model of allergic lung inflammation. Clinical data: efficacious and safe in an antihistamine-resistant chronic spontaneous urticaria.	[55,56]
resolvinE3 (RvE3),18-deoxy-RvE3	antagonism of BLT1	Decreased immune cell infiltration in a model of allergic inflammation, improvement in lung function parameters. Decreased polymorphonuclear leukocytes (PMNs) in the peritoneal exudates.	[57,58]
protectin D1	proposed action on GPR37	Decreased eosinophil recruitment to the lungs, improvement in lung function parameters and airway mucus secretion. Accelerated resolution of airway inflammation.	[59,60]
stable LXA_4_ analogues	ALX/FPR2 receptor agonist	Preclinical: Inhibition of airway hyperresponsiveness and inflammation; Clinical: well tolerated, topical- reduces the severity of infantile eczema, inhaled- improves lung function in mild asthmatics	[61,62,63]
S-777469	CB_2_ receptor agonist	Preclinical: inhibition of skin inflammation and eosinophil infiltration in mice by blocking the activities of 2-AG. Phase Ib/IIa clinical study for atopic dermatitis ongoing (NCT00697710).	[64]
JZL184	MGL inhibitor	Inhibition of eosinophil migration induced by 2-AG in vitro. Inhibition of airway hyperreactivity and airway inflammation in an LPS-induced model of airway inflammation.	[65,66]
GWJ-23	LysoPLD (autotaxin) inhibitor	Reduced inflammation in a triple-allergen mouse asthma model	[67]
ki16425	antagonist of LPA receptor 1 and 3	Prevents bradykinin induced asthmatic bronchoconstriction and reduces associated respiratory distress following allergen exposure	[68]
FPS-ZM1	RAGE receptor antagonist	Inhibits rIL-5/13–mediated airway mucus metaplasia and allergic inflammation, including eosinophil infiltration.	[69]
zileuton	5-LO inhibitor	In use: prophylaxis and treatment of chronic asthma	[70]
montelukast zafirlukast	Cys-LT_1_ receptor antagonist	In use: prophylaxis and chronic treatment of asthma, the prevention of exercise-induced bronchospasm, and the relief of symptoms of allergic rhinitis.	[71]
TAS-205 HQL-79	hPGDS inhibitor	In combination with montelukast showed additive inhibitory effects on eosinophil infiltration and late phase nasal obstruction in a model of allergic rhinitis. Ameliorated airway inflammation in an ovalbumin model of allergic airway inflammation.	[72,73]
fevipiprant	DP2 receptor antagonist	Phase III clinical studies did not show a statistically significant reduction in asthma exacerbations after adjusting for multiple testing; however, consistent and modest reductions in exacerbations rates were observed with the 450 mg (higher) dose.	[74]
ONO-0453	DP1 receptor antagonist	Shows improvement in all nasal and eye symptoms in a Phase II clinical study in seasonal allergic rhinitis patients.	[75]
butaprost	EP2 receptor agonist	Ameliorates airway inflammation and allergen-induced accumulation of eosinophils in the lungs of ovalbumin-sensitized mice. Limits airway hyperresponsiveness and inflammation in a house dust mite induced model.	[76,77]
iloprost	stable analogue of PGI_2_	Inhibits eosinophil trafficking and improves cardinal features of asthma in preclinical models	[78,79]
S-Y048	5-oxo-ETE receptor antagonist	Inhibits eosinophil infiltration in the skin following intradermal allergen challenge in primates.	[80]

**Table 2 ijms-22-04356-t002:** Effects of phospholipid cleavage products on eosinophil functions.

Bioactive Lipid Mediator	Mode of Action on Eosinophils	References
2-arachidonolyglycerol	Induces chemotaxis Induces calcium flux	[68,101,102] [103]
5-oxo-ETE	Increases chemotaxis Increases calcium flux, actin reorganization and ROS production Increases infiltration in the skin following intradermal application Upregulates eosinophil cationic protein in nasal polyps in vitro	[104,105,106] [104] [107] [108]
Cysteinyl leukotrienes	Induce chemotaxis and adhesion molecule upregulation Induce eosinophil trans endothelial migration, superoxide production and degranulation Elicit secretion from within cell-free human eosinophil granules	[109] [110] [111]
Leukotriene B4	Induces infiltration to the inflamed peritoneum	[112]
Lipoxin A4	Inhibits eicosanoid-induced tissue migration and airway hyperresponsiveness	[61]
Lysophosphatidic Acid	Increases CD11b upregulation, superoxide production, chemotaxis, calcium flux and actin reorganization Increases airway infiltration and superoxide production	[113] [114]
Lysophosphatidylcholine	Pro-inflammatory Induces CD11b upregulation, adhesion and calcium flux Increases infiltration in the airways and airway resistance Anti-inflammatory Inhibits CD11b upregulation, calcium flux, Akt and Erk phosphorylation as well as in vitro and in vivo migration	[115] [116] [101]
Lysophosphatidylserine	Induces degranulation Induces eosinophil extracellular trap formation	[102] [117]
Platelet Activating Factor	Induces calcium flux Increases chemokinesis and chemotaxis Induces degranulation Induces prostanoid secretion	[92] [93] [90] [91]
Prostaglandin D_2_	DP2 induces calcium flux, CD11b upregulation, migration and degranulation primes eosinophils for other chemoattractants DP1 prolongs survival enhances DP2 signaling	[118,119] [120,121] [118,122] [122,123]
Prostaglandin E_2_	Inhibits chemotaxis, CD11b upregulation, degranulation, calcium flux, cytoskeletal rearrangement and reactive oxygen production Prolongs their survival Decreases sputum counts	[76,124] [125,126] [127]
Prostacyclin	Inhibits migration and adhesion in vitro, reduces CD11b upregulation and activation Inhibits migration in vivo	[128] [79]
Protectin D1	Inhibits chemotaxis as well as CD11b and L-selectin expression Decreases airway infiltration and airway hyperresponsiveness	[129] [60]
Resolvin E3	Reduces infiltrating airway eosinophils in an HDM-induced airway inflammation model	[57]

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
