# Peer review of "Emerging Role of Phospholipase-Derived Cleavage Products in Regulating Eosinophil Activity: Focus on Lysophospholipids, Polyunsaturated Fatty Acids and Eicosanoids"

_ijms, 2021, doi:10.3390/ijms22094356_

Round 1
Reviewer 1 Report
This manuscript deals with an important, but not well-recognized topic, so it deserves to be published. To make it more attractive for the broader audience the authors should give some accent also on eosinophilia in cancer patients and on the relevance of non-enzymatic lipid peroxidation, which is of highest importance for physiology and pathophysiology of granulocytes in general.
Author Response
This manuscript deals with an important, but not well-recognized topic, so it deserves to be published. To make it more attractive for the broader audience the authors should give some accent also on eosinophilia in cancer patients and on the relevance of non-enzymatic lipid peroxidation, which is of highest importance for physiology and pathophysiology of granulocytes in general.
We thank the reviewer for the positive assessment of our manuscript. As per suggestion, we have included information on eosinophilia in different cancer subtypes available in the recent paper published by Grisaru-Tal et al. [1] (Lines 45-48). Moreover, a separate section on non-enzymatic lipid peroxidation has been added. Therein we described eosinophils as important ROS producers as well as delineated the potential roles of non-enzymatic lipid peroxidation products in asthma and allergic diseases (lines -624-667).
Reviewer 2 Report
The manuscript perfectly fits the International Journal of Molecular Sciences and will certainly attract attention of even broader audience, especially because it deals with very important topics in a very modern way. Therefore, it deserves considering for publication, but after some major and minor revision, which should include
MAJOR ISSUES
- English language, including style, should be revised.
- Are there any data about relationship between eosinophils and lysosomal PLA2 (LPLA2)?
- Line 294-318 you mentioned about eicosanoids but you have a separate section,you should organize your chapter.
- Should be written more about endocannabinoids.Maybe you write a separate chapter.
- In the section on eicosanoids, only leukotrienes and prostaglandins are described. Is there any research on other eicosanoids?
|MINOR CORRECTIONS
- Line 10 Should you write about concentration when talking about an enzyme (better expression)?
- Line 65 I am not sure if this sentence is needed ,,One of such pathways has previously already been successfully targeted in allergy with the development of the cysteinyl leukotriene receptor 1 (CysLT1) antagonist montelukast’’ If so, please insert literature.
- Line 68 In the aim of study, should be mention about lipid mediators, eicosanoids, which the entire chapter was wriiten.
- both chapters have the same number 1.4 Phospholipase A2 cleavage products: free fatty acids ten sam numer co 2.1.4. Phospholipase A2 cleavage products: lysophospholipids
Author Response
The manuscript perfectly fits the International Journal of Molecular Sciences and will certainly attract attention of even broader audience, especially because it deals with very important topics in a very modern way. Therefore, it deserves considering for publication, but after some major and minor revision, which should include:
MAJOR ISSUES
- English language, including style, should be revised.
We have thoroughly revised the language and style of the manuscript. For clarity reasons, we did not mark all grammatical and stylistic changes in the document.
- Are there any data about relationship between eosinophils and lysosomal PLA2 (LPLA2)?
As of today, no studies exist about the relationship between lysosomal PLA2 and eosinophils. LPLA2 has been found to be highly expressed by alveolar macrophages and was reported to contribute to airway surfactant degradation [2]. Since our review primarily focuses on eosinophil targeting, mentioning this LPLA2 function would be out of the scope of the review.
- Line 294-318 you mentioned about eicosanoids but you have a separate section, you should organize your chapter.
Thank you for suggesting an improved way of structuring our paper. We have now revised the order of the paper. All information on eicosanoids has now been included in chapter 3. Only direct effects of free fatty acids are mentioned in chapter 2.1.5.
- Should be written more about endocannabinoids. Maybe you write a separate chapter.
According to the suggestion, the endocannabinoid section was expanded and transformed into a separate chapter for clarity reasons (lines 543-575). However, due to the complexity of the topic, we decided to only include the studies directly related to eosinophil research. The reader is referred to a recent comprehensive review of the endocannabinoid research in allergic diseases by Angelina et al. (Collegium Internationale Alergologicum 2020 Update) [3].
- In the section on eicosanoids, only leukotrienes and prostaglandins are described. Is there any research on other eicosanoids?
Thank you for your comment. We have included prostaglandins (including prostacyclin), leukotrienes and 5-oxo-ETE. According to your suggestion we added some additional information on 5-HETE (lines 489-494) as well as on eoxins (lines 372-383). Since we have not identified any recent advances in connection with thromboxane targeting and eosinophils, we did not include it in the manuscript.
|MINOR CORRECTIONS
- Line 10 Should you write about concentration when talking about an enzyme (better expression)?
Thank you for pointing out our error. Indeed, the concentration, but also the activity and not activation of the enzymes is increased. We have corrected the manuscript accordingly (line 10).
- Line 65 I am not sure if this sentence is needed ,,One of such pathways has previously already been successfully targeted in allergy with the development of the cysteinyl leukotriene receptor 1 (CysLT1) antagonist montelukast’’ If so, please insert literature.
Thank you for your suggestion. We have removed this sentence from the manuscript.
- Line 68 In the aim of study, should be mention about lipid mediators, eicosanoids, which the entire chapter was writen.
Thank you for pointing out this issue. We have added eicosanoids to the aim of the study (Line 69).
- both chapters have the same number 1.4 Phospholipase A2 cleavage products: free fatty acids ten sam numer co 2.1.4. Phospholipase A2 cleavage products: lysophospholipids
Thank you, we have amended the chapter numbers.
Reviewer 3 Report
Suggestions for Authors:
- Line 72-74 of the manuscript. “Phospholipases are known allergens in insect sting reactions; for example, 72 phospholipase A2 (Api m 1) was identified as a major allergen of honeybee venom, with 73 approximately 97% of patients with bee venom allergy developing IgE reactive antibodies 74 against it [30,31] ” is out of context for the manuscript, as it introduces Phospholipases as a major proteinaceous allergen in the bee venoms and it should be skipped and led by the enzymatic functions and generation of bioactive lipids.
- Line 111-116, emphasizing the role of macrophage-associated sPLA2-V in governing lung type 2 response is missing a key reference. “Macrophages regulate lung ILC2 activation via Pla2g5-dependent mechanisms. Mucosal Immunologyvolume 11, pages615–626(2018)”
- Line 268-278, describing the importance of an ω-3 and ω-6 PUFA balance 268 in asthma and allergy can be avoided, as it takes the focus away from the bioactive mediators and eosinophils and adds unnecessary length to the manuscript. Effects of dietary ω-6 versus ω-3 supplementation on asthma outcomes are certainly important but not needed here.
- To greatly enhance the impact and reach of the paper, a table should be included with comprehensively describing published effects/functions of each phospholipase-derived cleavage product specifically in/thru eosinophils. This will improve the readability of the paper and deliver the message in a concise manner.
Author Response
- Line 72-74 of the manuscript. “Phospholipases are known allergens in insect sting reactions; for example, 72 phospholipase A2 (Api m 1) was identified as a major allergen of honeybee venom, with 73 approximately 97% of patients with bee venom allergy developing IgE reactive antibodies 74 against it [30,31] ” is out of context for the manuscript, as it introduces Phospholipases as a major proteinaceous allergen in the bee venoms and it should be skipped and led by the enzymatic functions and generation of bioactive lipids.
Thank you for the suggestion. Indeed, this part is out of scope of the manuscript. We have removed this part from the manuscript.
- Line 111-116, emphasizing the role of macrophage-associated sPLA2-V in governing lung type 2 response is missing a key reference. “Macrophages regulate lung ILC2 activation via Pla2g5-dependent mechanisms. Mucosal Immunologyvolume 11, pages615–626(2018)”
Thank you for pointing out our error. We have added the missing reference to the manuscript (Line 110).
- Line 268-278, describing the importance of an ω-3 and ω-6 PUFA balance 268 in asthma and allergy can be avoided, as it takes the focus away from the bioactive mediators and eosinophils and adds unnecessary length to the manuscript. Effects of dietary ω-6 versus ω-3 supplementation on asthma outcomes are certainly important but not needed here.
Thank you for your suggestion, we have removed this part of the manuscript.
- To greatly enhance the impact and reach of the paper, a table should be included with comprehensively describing published effects/functions of each phospholipase-derived cleavage product specifically in/thru eosinophils. This will improve the readability of the paper and deliver the message in a concise manner.
We agree with your recommendation; therefore, we included a synopsis table (Table 2) of all bioactive lipid mediators and their published effects on eosinophils.
Round 2
Reviewer 2 Report
I agree with your comments